# Genetic Association of *PCSK5* and *MUC2* Gene Polymorphisms with Recurrent Pregnancy Loss (RPL)

**DOI:** 10.3390/ijms26146585

**Published:** 2025-07-09

**Authors:** Chang Soo Ryu, Ji Hyang Kim, Eun Ju Ko, Hyeon Woo Park, Jae Hyun Lee, Ji Eun Shin, Young Ran Kim, Eun Hee Ahn, Nam Keun Kim

**Affiliations:** 1Department of Life Science, Graduate School, CHA University, 335 Pangyo-ro, Bundang-gu, Seongnam 13488, Republic of Korea; regis2040@nate.com (C.S.R.); ejko05@naver.com (E.J.K.); aabb1114@naver.com (H.W.P.); athe7a@naver.com (J.H.L.); 2Division of Life Sciences, College of Life Sciences, CHA University, 335 Pangyo-ro, Bundang-gu, Seongnam 13488, Republic of Korea; 3Department of Obstetrics and Gynecology, CHA Bundang Medical Center, School of Medicine, CHA University, 59 Yatap-ro, Bundang-gu, Seongnam 13496, Republic of Korea; bin0902@chamc.co.kr (J.H.K.); 1219annie@cha.ac.kr (J.E.S.); happyiran@cha.ac.kr (Y.R.K.)

**Keywords:** recurrent pregnancy loss, whole-exome sequencing, genetic variant, biomarker

## Abstract

Recurrent pregnancy loss (RPL) is defined as the occurrence of two or more consecutive pregnancy losses before 20 weeks of gestation, encompassing both embryonic and fetal losses. Although previous studies have provided substantial insights into RPL, the causes in many cases remain unexplained. This lack of information has prompted continued investigation into various risk factors, including those identified through next-generation sequencing (NGS). In the present study, whole-exome sequencing (WES) was used to identify genes potentially associated with RPL and infertility, which may serve as novel biomarkers. Confirmation of the association between these genetic variants and RPL may help to develop functional biomarkers for early diagnosis. The findings revealed that the *PCSK5* rs1110222 G > A polymorphism was significantly associated with a reduced risk of RPL. In contrast, the *MUC2* rs10902088 C > T polymorphism was associated with an increased risk of RPL among women with more than four pregnancy losses. Notably, the A-T allele combination of *PCSK5* rs1110222 G > A and *MUC2* rs10902088 C > T showed a significant association with a decreased risk of RPL relative to the G-C combination. In conclusion, this study confirms that the *PCSK5* rs1110222 G > A and *MUC2* rs10902088 C > T polymorphisms are genetically associated with the prevalence of RPL in Korean women.

## 1. Introduction

Recurrent pregnancy loss (RPL) is defined as the occurrence of two or more consecutive pregnancy losses before 20 weeks of gestation, encompassing both embryonic and fetal losses. RPL affects approximately 5% of couples of reproductive age; it represents an important global challenge in contemporary obstetrics and reproductive medicine [1,2]. Although previous basic and clinical studies have yielded important findings, the underlying causes of many RPL cases remain unexplained [2]. This uncertainty has prompted investigation of additional risk factors through various approaches, including next-generation sequencing (NGS) [3,4,5,6]. Several studies utilizing whole-exome sequencing (WES) have explored biomarkers of RPL; however, most have focused on non-Korean populations [5,6,7]. Recent research has demonstrated the potential to detect pathogenic and exonic RPL-associated variants using NGS [8,9].

Advances in NGS have substantially enhanced efforts to identify genes and single-nucleotide polymorphisms (SNPs) associated with pregnancy loss. WES is commonly used to identify genetic regions associated with reproductive disorders [5]. This method has become a widely used tool for identifying genetic regions linked to reproductive disorders because it is more cost-effective, easier to interpret, and yields more manageable datasets than whole-genome sequencing [10,11]. Consequently, many studies have identified causative genes and potential biomarkers of reproductive diseases via WES [12]. Interestingly, endometriosis-associated genes and SNPs have also been detected through WES analyses of RPL cases [13].

Endometriosis is an estrogen-dependent chronic disease that causes infertility [14,15]. Several studies have shown an association between endometriosis and pregnancy loss, including RPL, in affected women [16,17,18]. Additional research has demonstrated associations between polymorphic biomarkers and RPL in patients with endometriosis [19,20]. Other studies have indicated the potential for shared biomarkers between endometriosis and infertility [21,22,23]. However, the associations of specific gene polymorphisms implicated in endometriosis with RPL have not been clearly demonstrated. This study aimed to investigate associations between endometriosis-related gene polymorphisms and RPL through WES. Comparative analysis of WES data from RPL patients and healthy controls identified polymorphisms in the proprotein convertase subtilisin/kexin type 5 (*PCSK5*) and mucin 2 (*MUC2*) genes, both previously reported in association with endometriosis [13,24]. The *PCSK5* gene mediates posttranslational endoproteolytic processing in several integrin alpha subunits; it also reportedly influences ovarian follicle development in mouse models [25,26]. Moreover, the expression of proprotein convertase substrates has been identified through genome-wide expression correlation analysis [27]. We previously reported a potential association between mucin 4 (*MUC4*)—a member of the same mucin family as *MUC2*—and reproductive diseases such as RPL, demonstrated by biomarkers [9,24,28,29]. Other studies have also indicated that the *MUC2* gene plays a prominent role in maintaining pregnancy health and supporting fetal development [30].

Using WES, we identified several genes associated with RPL and infertility that may serve as novel biomarkers of reproductive diseases. Efforts to confirm the associations between these newly identified genetic variants and RPL may facilitate development of functional biomarkers for early diagnosis.

## 2. Results

### 2.1. Comparison of Baseline Clinical Profiles

Table 1 presents selected clinical characteristics of RPL patients and control participants. Multiple variables significantly differed between the two groups. In particular, hematocrit, luteinizing hormone, blood urea nitrogen, and high-density lipoprotein (HDL) levels were significantly higher in the RPL group than among the controls (*p* = 0.008, *p* < 0.0001, *p* = 0.0004, and *p* = 0.024, respectively). Additional clinical parameters—including follicle-stimulating hormone, prothrombin time, homocysteine, and creatinine levels—also showed significant differences between the groups.

### 2.2. Genotype Frequency Analyses of PCSK5 and MUC2 Gene Polymorphisms in RPL Patients, RPL Patient Subgroups, and Control Participants

After clinical profile comparisons, we analyzed the genotype frequencies of *PCSK5* and *MUC2* gene polymorphisms among RPL patients, RPL patient subgroups, and control participants (Table 2). RPL patients were further classified into subgroups based on the number of pregnancy losses, with categories defined as pregnancy losses ≥ 3 and pregnancy losses ≥ 4. The *PCSK5* polymorphisms rs1110222 G > A and rs2259969 A > G, as well as the *MUC2* polymorphisms rs7103978 A > G and rs10902088 C > T, were examined in all groups. We identified a significant association between *PCSK5* rs1110222 G > A and RPL risk. The *PCSK5* rs1110222 GA genotype was significantly associated with a reduced risk of RPL, as shown in Table 2 (*p* = 0.043). Genotype frequencies for all polymorphisms were consistent with Hardy–Weinberg equilibrium (*p* > 0.05). We also identified a significant association between *MUC2* rs10902088 C > T and RPL risk in the pregnancy losses ≥ 4 subgroup. Under the recessive model, the *MUC2* rs10902088 C > T polymorphism was significantly associated with an increased risk of RPL (Table 2, *p* = 0.012). Although trends were observed for other variants (e.g., the *PCSK5* rs1110222 GA genotype and *MUC2* rs10902088 TT genotype), these were not statistically significant.

### 2.3. Combination Analyses of PCSK5 and MUC2 Gene Polymorphisms Between RPL Patients and Control Participants

Next, we compared allele combinations among RPL patients and control participants (Table 3). Analysis of the four polymorphisms identified seven allele combinations significantly associated with RPL risk. Notably, combinations involving the *MUC2* rs7103978 G allele were associated with a decreased risk of RPL relative to those containing the A allele. For instance, the G-A-G-C combination (*PCSK5* rs1110222 G > A/*PCSK5* rs2259969 A > G/*MUC2* rs7103978 A > G/*MUC2* rs10902088 C > T) had an odds ratio (OR) of 0.043 (95% confidence interval [CI] = 0.003–0.746, *p* = 0.001). Similarly, the G-G-C combination (*PCSK5* rs1110222 G > A/*MUC2* rs7103978 A > G/*MUC2* rs10902088 C > T) had an OR of 0.146 (95% CI = 0.032–0.674, *p* = 0.006), and the G-C combination (*MUC2* rs7103978 A > G/*MUC2* rs10902088 C > T) had an OR of 0.217 (95% CI = 0.070–0.672, *p* = 0.005). Furthermore, combinations containing the *PCSK5* rs1110222 A allele were consistently associated with a reduced risk of RPL relative to those containing the G allele. For example, the allele combinations A-A-A-T (*PCSK5* rs1110222 G > A/*PCSK5* rs2259969 A > G/*MUC2* rs7103978 A > G/*MUC2* rs10902088 C > T; OR = 0.285, 95% CI = 0.129–0.627, *p* = 0.0003), A-T (*PCSK5* rs1110222 G > A/*MUC2* rs10902088 C > T; OR = 0.631, 95% CI = 0.402–0.990, *p* = 0.044), and A-A (*PCSK5* rs1110222 G > A/*PCSK5* rs2259969 A > G; OR = 0.455, 95% CI = 0.264–0.786, *p* = 0.004) were associated with a reduced risk of RPL.

Genotype combination analyses (Table 4) further supported these findings. Notably, the GA/AA genotype combination for *PCSK5* rs1110222 G > A/*PCSK5* rs2259969 A > G was significantly associated with a reduced risk (OR = 0.335, 95% CI = 0.169–0.664, *p* = 0.002), as was the GA/AG combination for *PCSK5* rs1110222 G > A/*MUC2* rs7103978 A > G (OR = 0.653, 95% CI = 0.427–0.998, *p* = 0.049).

### 2.4. Analysis of Variance Involving PCSK5 and MUC2 Polymorphisms with Various Clinical Factors

We assessed several clinical parameters, including activated partial thromboplastin time (aPTT), the proportions of CD4⁺ helper T cells and CD56⁺ natural killer cells, and HDL levels. The *PCSK5* rs1110222 AA genotype and A allele groups exhibited an increased aPTT compared with the GG genotype and G allele groups (Figure 1). Similarly, the *PCSK5* rs2259969 GG genotype and G allele groups showed a higher aPTT than the AA genotype and A allele groups (Figure 2). For the *MUC2* rs7103978 A > G polymorphism, individuals with the AG genotype and G allele groups had a higher proportion of CD4⁺ helper T cells than those with the AA genotype and A allele groups (Figure 3), whereas the proportion of CD56⁺ natural killer cells was lower in the AG genotype and G allele groups than in the AA genotype and A allele groups (Figure 4).

## 3. Discussion

In this study, WES was used to confirm the association between gene polymorphisms related to endometriosis and RPL. The findings showed significant associations between *PCSK5* and *MUC2* gene polymorphisms and RPL risk. Specifically, the *PCSK5* rs1110222 G > A polymorphism was associated with a significantly reduced risk of RPL, whereas the *MUC2* rs10902088 C > T polymorphism was linked to an increased risk, particularly among women with ≥4 pregnancy losses. Additionally, allele combination analyses indicated that the *MUC2* rs7103978 G allele was associated with a decreased risk of RPL relative to the A allele. The combination of *PCSK5* rs1110222 G > A and *MUC2* rs10902088 C > T (A-T combination) was significantly associated with a lower RPL risk relative to the G-C combination.

Although the *PCSK5* and *MUC2* genes have previously been implicated in endometriosis, their involvement in pregnancy loss has not been well characterized [13,24]. Therefore, the present study provides foundational evidence linking these gene polymorphisms to RPL risk and suggests that they may serve as novel biomarkers for early diagnosis. As previously noted, the *PCSK5* gene encodes a protein responsible for proprotein processing and plays a role in female pregnancy, particularly in embryo implantation [25,31,32]. The *MUC2* gene encodes a protein that forms a gel-like mucus layer in the gastrointestinal tract, protecting the intestinal lining [28,30]. Recent bioinformatic analyses have identified *PCSK5* as a potential biomarker of RPL [33]. Additionally, altered *MUC2* gene expression has been observed in the endometrium of patients with recurrent implantation failure [34]. Polymorphisms in *MUC2* have also been associated with the development of endometriosis and infertility [24]. However, to our knowledge, no previous research has established an association between these genes and RPL risk. Our findings highlight the potential roles of *PCSK5* and *MUC2* gene polymorphisms as biomarkers of RPL risk.

However, this study has some limitations. First, the functional mechanisms by which the identified polymorphisms in *PCSK5* and *MUC2* contribute to the pathogenesis of RPL remain unclear because we did not perform functional analyses of these SNPs. Second, we did not assess the immunologic profiles and expression levels of the *PCSK5* and *MUC2* genes assessed. Third, the study population was limited to Korean women, which may restrict the generalizability of the findings. Further epidemiological and functional studies are needed to confirm and expand upon these results. Fourth, the group of WES participants comprised only 45 women, so we will progress further by applying WES to a larger subset to improve statistical power and detect rare variants. Fifth, we had no adjusting confounding variables, so we will plan to investigate more specific confounding variables and attempt to adapt our analysis of the results. Despite the application of strict inclusion criteria and standardized measurement protocols, residual confounding cannot be fully excluded, particularly regarding comorbid conditions and treatment effects. Future studies will include multivariable adjustment for clinical covariates to address this limitation. In conclusion, this study confirmed significant genetic associations between the *PCSK5* rs1110222 G > A and *MUC2* rs10902088 C > T polymorphisms and the prevalence of RPL in Korean women. Moreover, allele and genotype combination analyses revealed significant differences between RPL patients and control participants. Our findings underscore the potential of these gene polymorphisms as biomarkers of RPL and highlight the need for further investigation into their roles in the condition’s pathogenesis.

## 4. Materials and Methods

### 4.1. Study Approval and Population Used in This Study

This study was approved by the CHA Bundang Medical Center Institutional Review Board (IRB No. BD2010-123D), and all protocols were conducted in accordance with the Declaration of Helsinki. Written informed consent was obtained from all participants. Furthermore, this study included a total of 574 participants. The control group consisted of 236 women recruited from CHA Bundang Medical Center. Controls exhibited a normal 46,XX karyotype, regular menstrual cycles, a history of at least one naturally conceived pregnancy, and no history of pregnancy loss (PL). The patient group included 338 women diagnosed with recurrent pregnancy loss (RPL) at the Department of Obstetrics and Gynecology, Bundang CHA Medical Center, CHA University (Seongnam, Republic of Korea), between March 1999 and December 2012. RPL was defined as the occurrence of two or more consecutive pregnancy losses, with each pregnancy failure confirmed by human chorionic gonadotropin testing, ultrasound, or physical examination before 20 weeks of gestation. Patients were further categorized into subgroups based on the number of pregnancy losses. Individuals whose pregnancy loss was attributable to factors such as anatomical, hormonal, chromosomal, infectious, autoimmune, or thrombotic causes were excluded. Anatomical abnormalities were identified via sonography, hysterosalpingogram, hysteroscopy, computed tomography, or magnetic resonance imaging.

### 4.2. Estimation of Biochemical Factor Concentrations

Blood samples were collected from all participants in anticoagulant tubes after a 12 h fasting period. Participants with comorbidities such as diabetes, hypertension, or dyslipidemia were included only if their conditions were clinically well-controlled and stable; individuals with acute or uncontrolled metabolic disease were excluded. These measures were taken to reduce potential confounding due to treatment-related effects. Plasma was separated by centrifugation at 1000× *g* for 15 min. Uric acid and total cholesterol levels were measured by enzymatic colorimetry (Roche Diagnostics, GmbH, Mannheim, Germany). Levels of high-density lipoprotein cholesterol were measured by enzymatic colorimetry using a set of commercial reagents (TBA 200FR NEO, Toshiba Medical Systems, Tochigi, Japan). Homocysteine levels were quantified by fluorescence polarization immunoassays using an Abbott IMx analyzer (Abbott Laboratories, Abbott Park, IL, USA). Folate and creatinine concentrations were determined using a competitive immunoassay with the ACS 180 Plus automated chemiluminescence system (Bayer Diagnostics, Tarrytown, NY, USA). Complete blood counts, including white blood cells, red blood cells, hemoglobin, and platelet counts, were obtained using the Sysmex XE 2100 automated hematology system (Sysmex Corporation, Kobe, Japan). Prothrombin time and activated partial thromboplastin time were measured using an automated photo-optical coagulometer (ACL TOP; Mitsubishi Chemical Medicence, Tokyo, Japan).

### 4.3. Flow Cytometry Analysis of Immune Cell Proportions

Immune cell proportions were analyzed by flow cytometry using CellQuest software version 5.1 (BD FACS Calibur; BD Biosciences, NJ, USA). Peripheral blood mononuclear cells (2.5 × 10^5^) were stained for 30 min at 4 °C in the dark, washed twice with 2% phosphate-buffered saline containing 1% bovine serum albumin and 0.01% sodium azide (i.e., FACS wash buffer), and fixed with 1% formaldehyde (Sigma-Aldrich, St. Louis, MO, USA). Fluorescently labeled monoclonal antibodies (labeled with fluorescein isothiocyanate, phycoerythrin, peridinin chlorophyll protein, or allophycocyanin) specific for CD3, CD4, CD8, CD19, CD16, and CD56 were used at a dilution of 1:1000.

### 4.4. Hormone Assays

Female hormone levels (follicle-stimulating hormone, estradiol (E2), luteinizing hormone) were measured on day 2 or 3 of the menstrual cycle when hormonal fluctuations are minimal and E2 interference is low, to ensure stable and accurate measurement of ovarian reserve and reproductive potential [35]. E2, thyroid-stimulating hormone (TSH), and prolactin levels were measured via radioimmunoassay (Beckman Coulter, Brea, CA, USA). Follicle-stimulating hormone and luteinizing hormone levels were measured by enzyme-linked immunosorbent assays (Siemens, Munich, Germany).

### 4.5. Genotyping

Genomic DNA was extracted from peripheral blood samples using a G-DEX(TM) Genomic DNA Extraction Kit for blood (iNtRON Biotechnology, Seongnam, Republic of Korea). We examined the following polymorphisms in this study: *PCSK5* rs1110222 G > A (PCR-RFLP), *PCSK5* rs2259969A > G (real-time PCR), *MUC2* rs7103978 A > G (real-time PCR), and *MUC2* rs10902088 C > T (real-time PCR). The PCR primers for each polymorphism were as follows: *PCSK5* rs1110222 G > A (forward 5′-GAC CTA ATT CCT TTT CCC CAG-3′/reverse 5′- GCT ACA GCG TTC ACA TAG G-3′), *PCSK5* rs2259969 A > G (forward 5′-ATT GCC ACA AGT CCT GCT-3′/reverse 5′-ATG CAG CTC CCA CGA G-3′), *MUC2* rs7103978 A > G (forward 5′-ACG TGG GCT TAG GTA CCA GGA CT-3′/reverse 5′-A GTG TGG CTG CTA TGT CGA GGA-3′), and *MUC2* rs10902088 C > T (forward 5′-TAT GAG CCA TGT GGG AAC-3′/reverse 5′-TGC TCA CCC TCC AGG TA-3′). For the *PCSK5* rs1110222 G > A polymorphism, PCR products were digested with *Hae*III restriction enzymes (New England Bio Laboratories, Ipswich, MA, USA) at 37 °C for 16 h. For quality control, selected PCR products for each polymorphism were duplicated and confirmed by DNA sequencing using an ABI 3730xl DNA Analyzer (Applied Biosystems, Foster City, CA, USA), yielding a 100% concurrence in sample quality.

### 4.6. Whole-Exome Sequencing (WES) Analysis

Whole-exome sequencing was conducted on 45 women using the Illumina HiSeq 2000 platform (Macrogen, Seoul, Republic of Korea). Exome enrichment was performed with the SeqCap EZ HGSC VCRome kit (Roche NimbleGen, Pleasanton, CA, USA), which targets 189,028 non-overlapping exons in 23,585 human genes (total target size: 45.1 Mb). Paired-end sequencing was performed at the Human Genome Sequencing Center (HGSC). Raw sequence reads were aligned to the reference human genome (hg19/GRCh37), and variant calling was executed using the HGSC proprietary software “Atlas2” (https://www.hgsc.bcm.edu/software/atlas-2, accessed on 1 January 2020). Individual variant call format (VCF) files were generated and subsequently merged using VCF tools. Variants that failed to meet quality control criteria were excluded; these criteria included an SNV posterior probability of ≥0.95, a minimum of 3 reads, a variant-to-read ratio of ≥0.1, the detection of variant reads on both strands, and a total coverage of ≥6. Minor allele frequencies (MAFs) were obtained from the dbSNP (https://www.ncbi.nlm.nih.gov/snp/, accessed on 1 January 2020) and the 1000 Genomes Project phase 3 populations.

### 4.7. Statistical Analysis

Differences in the frequencies of *PCSK5* and *MUC2* polymorphisms between patients and controls were evaluated using Fisher’s exact test and logistic regression. Associations between the polymorphisms and RPL incidence were estimated using adjusted odds ratios (AORs) with 95% confidence intervals (CIs), with adjustments made for age. These AORs and 95% CIs were also used to assess the relationships between specific polymorphisms and allele combinations. A *p*-value of < 0.05 was considered statistically significant. All polymorphisms were confirmed to be in Hardy–Weinberg equilibrium (*p* > 0.05). Statistical analyses were performed using GraphPad Prism 4.0 (GraphPad Software, Inc., San Diego, CA, USA), StatsDirect version 2.4.4 (StatsDirect Ltd., Altrincham, UK), HaploView 4.1 (Broad Institute of MIT and Harvard, Boston, MA, USA), and HAPSTAT 3.0 (University of North Carolina, Chapel Hill, NC, USA). Additionally, gene–gene interaction analyses were conducted using the open-source multi-dimensional reduction (MDR) software package version 2.0 (www.epistasis.org, accessed on 1 January 2020), which assessed all possible combinations of polymorphisms to identify those with significant synergistic effects.

## Figures and Tables

**Figure 1 ijms-26-06585-f001:**
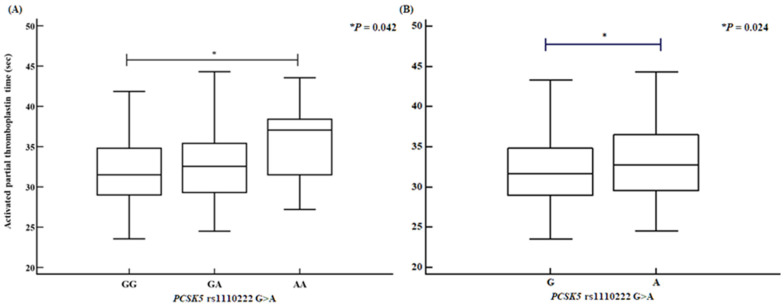
Activated partial thromboplastin time disparities of *PCSK5* rs1110222 G > A polymorphisms, determined by analysis of variance (ANOVA). *PCSK5* rs1110222 G > A polymorphisms confirm association with increased activated partial thromboplastin time [(**A**) GG vs. GA vs. AA: GG, 32.04 ± 4.01; GA, 32.77 ± 4.31; AA, 35.79 ± 5.71; *p* = 0.042. (**B**) G allele vs. A allele: G allele, 32.14 ± 4.05; A allele, 33.16 ± 4.56; *p* = 0.024].

**Figure 2 ijms-26-06585-f002:**
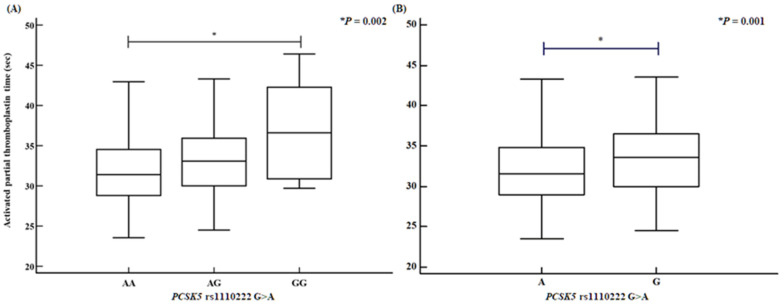
Activated partial thromboplastin time disparities of *PCSK5* rs2259969 A > G polymorphisms, determined by analysis of variance (ANOVA). *PCSK5* rs2259969 A > G polymorphisms confirm association with increased activated partial thromboplastin time [(**A**) AA vs. AG vs. GG: AA, 31.93 ± 4.06; AG, 32.94 ± 3.86; GG, 36.96 ± 6.41; *p* = 0.002. (**B**) A allele vs. G allele: A allele, 32.07 ± 4.04; G allele, 33.54 ± 4.48; *p* = 0.001].

**Figure 3 ijms-26-06585-f003:**
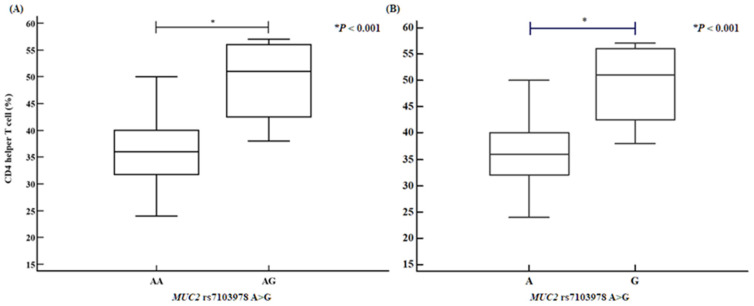
CD4 helper T cell proportion disparities of *MUC2* rs7103978 A > G polymorphisms, determined by analysis of variance (ANOVA) in RPL patients. *MUC2* rs7103978 A > G polymorphisms confirm association with increased CD4 helper T cell percentage [(**A**) AA vs. AG: AA, 35.81 ± 6.78; AG, 49.25 ± 8.66; *p* < 0.001. (**B**) A allele vs. G allele: A allele, 36.15 ± 7.10; G allele, 49.25 ± 8.66; *p* < 0.001; number of patients: 81].

**Figure 4 ijms-26-06585-f004:**
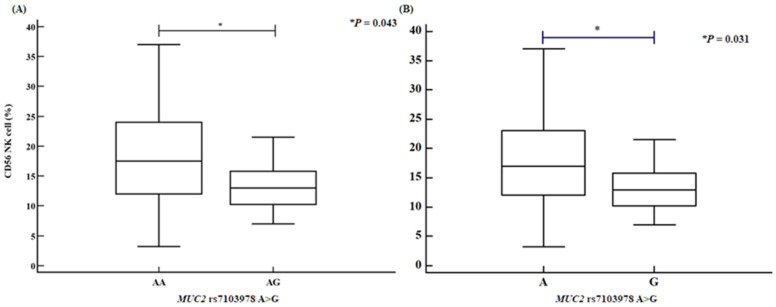
CD56 natural killer cell proportion disparities of *MUC2* rs7103978 A > G polymorphisms, determined by analysis of variance (ANOVA) in RPL patients. *MUC2* rs7103978 A > G polymorphisms confirm association with decreased CD56 natural killer cell proportion [(**A**) AA vs. AG: AA, 18.13 ± 7.60; AG, 14.08 ± 5.05; *p* = 0.043. (**B**) A allele vs. G allele: A allele, 17.90 ± 7.52; G allele, 14.08 ± 5.04; *p* = 0.031; number of patients: 139].

**Table 1 ijms-26-06585-t001:** Baseline characteristics between recurrent pregnancy loss patients and controls.

Characteristics	Controls (*n* = 236)	RPL Patients (*n* = 338)	*p* ^a^
Age (years, mean ± SD)	33.42 ± 5.84	33.32 ± 4.61	0.803
Previous pregnancy losses (*n*, %)	NA	3.32 ± 1.87	NA
Live births (*n*, %)	1.67 ± 0.73	NA	NA
Mean gestational age (weeks)	39.30 ± 1.67	NA	NA
BMI (kg/m^2^)	21.76 ± 3.32	21.62 ± 4.05	0.731
FSH (mIU/mL)	8.69 ± 4.80	7.69 ± 10.96	**<0.0001 ^b^**
E2 (pg/mL)	28.34 ± 28.63	34.63 ± 25.43	0.057
LH (mIU/mL)	3.79 ± 4.68	6.30 ± 12.46	**<0.0001 ^b^**
TSH (uIU/mL)	1.75 ± 0.90	2.20 ± 1.54	0.473 ^b^
Prolactin (ng/mL)	10.39 ± 12.76	16.03 ± 13.32	0.403
PT (sec)	11.17 ± 2.18	11.65 ± 0.91	**<0.0001 ^b^**
aPTT (sec)	32.12 ± 4.13	32.33 ± 4.15	0.674
Antithrombin (%)	98.86 ± 20.41	89.36 ± 26.54	0.427
Fibrinogen (mg/dL)	501.80 ± 101.17	366.70 ± 54.11	0.065
Homocysteine (umol/L)	10.03 ± 6.54	6.97 ± 1.97	**0.011 ^b^**
Folate (ng/mL)	11.64 ± 8.62	15.35 ± 16.89	0.308 ^b^
Glucose (mg/dL)	98.56 ± 22.10	97.55 ± 16.54	0.628 ^b^
BUN (mg/dL)	9.50 ± 3.49	10.87 ± 6.63	**0.0004 ^b^**
Creatinine (mg/dL)	0.65 ± 0.15	0.74 ± 0.12	**<0.0001 ^b^**
Uric acid (mg/dL)	3.97 ± 0.94	3.86 ± 0.91	0.429
Total cholesterol (mg/dL)	226.75 ± 58.27	180.57 ± 47.00	**<0.0001 ^b^**
Triglyceride (mg/dL)	195.06 ± 153.20	156.52 ± 128.87	0.187
LDL (mg/dL)	102.35 ± 28.08	103.00 ± 39.72	0.954
HDL (mg/dL)	57.09 ± 11.70	69.48 ± 19.14	**0.024 ^b^**
WBC (10^3^/uL)	6.45 ± 2.51	6.96 ± 2.45	**0.032**
RBC (10^6^/uL)	4.16 ± 0.40	4.19 ± 0.39	0.400
Hgb (g/dL)	12.42 ± 2.42	12.55 ± 1.16	0.077 ^b^
Hct (%)	36.57 ± 3.47	37.43 ± 3.31	**0.008**
PLT (10^3^/uL)	249.67 ± 65.10	251.20 ± 58.29	0.791
Seg (%)	69.34 ± 8.93	62.34 ± 12.34	**<0.0001 ^b^**
Lym (%)	22.53 ± 7.61	28.65 ± 10.62	**<0.0001 ^b^**
Mono (%)	6.15 ± 2.18	5.54 ± 2.04	**0.034**
Eo (%)	1.29 ± 0.88	2.00 ± 1.59	**0.0001 ^b^**
Baso (%)	0.47 ± 0.28	0.45 ± 0.32	0.587
Vit D (ng/mL)	8.75 ± 0.21	14.13 ± 8.87	0.245 ^b^
CD56 NK cell (%)	NA	17.70 ± 7.46	NA
CD3 pan T cell (%)	NA	66.94 ± 8.60	NA
CD4 helper T cell (%)	NA	36.48 ± 7.42	NA
CD8 suppressor T cell (%)	NA	27.46 ± 7.73	NA
CD19 B cell (%)	NA	12.35 ± 4.74	NA

*p* ^a^ were calculated using two-sided *t*-test for continuous variables. ^b^ Mann–Whitney test for continuous data. RPL, recurrent pregnancy loss; SD, standard deviation; BMI, body mass index; FSH, follicle-stimulating hormone; E2, estradiol-2; LH, luteinizing hormone; TSH, thyroid-stimulating hormone; PT, prothrombin time; aPTT, activated partial thromboplastin time; BUN, blood urea nitrogen; LDL, low-density lipoprotein; HDL, high-density lipoprotein; WBC, white blood cell; RBC, red blood cell; Hgb, hemoglobin; Hct, hematocrit; PLT, platelet; Seg, seg neutrophil; Lym, lymphocyte; Mono, monocyte; Eo, eosinophil; Baso, basophil; Vit D, vitamin D; NK, natural killer; NA, not applicable. *p*-values < 0.05 are bold.

**Table 2 ijms-26-06585-t002:** Genotype frequency of *PCSK5* and *MUC2* gene polymorphisms in RPL subtype patients and controls.

Genotypes	Controls (*n* = 236)	PL ≥ 2 (*n* = 338)	AOR (95% CI)	*p **	PL ≥ 3 (*n* = 128)	AOR (95% CI)	*p **	PL ≥ 4 (*n* = 68)	AOR (95% CI)	*p **
*PCSK5* rs1110222 G > A										
GG	166 (70.3)	259 (76.6)	1.000 (reference)		96 (75.0)	1.000 (reference)		54 (79.4)	1.000 (reference)	
GA	66 (28.0)	70 (20.7)	**0.668** **(0.453–0.987)**	**0.043**	28 (21.9)	0.737 (0.443–1.226)	0.240	11 (16.2)	0.517 (0.254–1.051)	0.068
AA	4 (1.7)	9 (2.7)	1.419 (0.429–4.695)	0.566	4 (3.1)	1.747 (0.427–7.155)	0.438	3 (4.4)	2.350 (0.508–10.861)	0.274
Dominant (GG vs. GA + AA)			0.712 (0.488–1.038)	0.078		0.796 (0.488–1.298)	0.360		0.623 (0.324–1.196)	0.155
Recessive (GG + GA vs. AA)			1.581 (0.480–5.204)	0.451		1.893 (0.465–7.708)	0.373		2.718 (0.592–12.482)	0.199
HWE-*P*	0.375	0.116								
*PCSK5* rs2259969 A > G										
AA	168 (71.2)	239 (70.7)	1.000 (reference)		95 (74.2)	1.000 (reference)		52 (76.5)	1.000 (reference)	
AG	62 (26.3)	86 (25.4)	0.960 (0.653–1.410)	0.835	31 (24.2)	0.897 (0.542–1.483)	0.671	14 (20.6)	0.746 (0.385–1.448)	0.387
GG	6 (2.5)	13 (3.8)	1.506 (0.560–4.051)	0.418	2 (1.6)	0.597 (0.118–3.019)	0.533	2 (2.9)	1.100 (0.215–5.627)	0.909
Dominant (AA vs. AG + GG)			1.007 (0.696–1.458)	0.969		0.871 (0.534–1.422)	0.581		0.779 (0.414–1.465)	0.438
Recessive (AA + AG vs. GG)			1.530 (0.572–4.090)	0.397		0.615 (0.122–3.094)	0.555		1.184 (0.233–6.012)	0.839
HWE-*P*	0.922	0.143								
*MUC2* rs7103978 A > G										
AA	205 (86.9)	302 (89.3)	1.000 (reference)		114 (89.1)	1.000 (reference)		61 (89.7)	1.000 (reference)	
AG	31 (13.1)	36 (10.7)	0.789 (0.473–1.318)	0.366	14 (10.9)	0.818 (0.418–1.602)	0.558	7 (10.3)	0.777 (0.325–1.858)	0.571
GG	0 (0.0)	0 (0.0)	NA	NA	0 (0.0)	NA	NA	0 (0.0)	NA	NA
Dominant (AA vs. AG + GG)			0.789 (0.473–1.318)	0.366		0.818 (0.418–1.602)	0.558		0.777 (0.325–1.858)	0.571
Recessive (AA + AG vs. GG)			NA	NA		NA	NA		NA	NA
HWE-*P*	0.280	0.301								
*MUC2* rs10902088 C > T										
CC	83 (35.2)	108 (32.0)	1.000 (reference)		43 (33.6)	1.000 (reference)		22 (32.4)	1.000 (reference)	
CT	113 (47.9)	166 (49.1)	1.130 (0.777–1.641)	0.523	56 (43.8)	0.955 (0.586–1.557)	0.854	25 (36.8)	0.834 (0.440–1.581)	0.579
TT	40 (16.9)	64 (18.9)	1.230 (0.755–2.003)	0.406	29 (22.7)	1.404 (0.768–2.569)	0.271	21 (30.9)	2.020 (0.992–4.113)	0.053
Dominant (CC vs. CT + TT)			1.152 (0.810–1.639)	0.430		1.070 (0.679–1.685)	0.771		1.136 (0.639–2.018)	0.664
Recessive (CC + CT vs. TT)			1.148 (0.743–1.775)	0.534		1.443 (0.844–2.467)	0.180		**2.211** **(1.192–4.101)**	**0.012**
HWE-*P*	0.884	0.988								

AOR, adjusted odds ratio; RPL, recurrent pregnancy loss; PCSK5, proprotein convertase subtilisin/kexin type 5; MUC2, mucin 2; HWE, Hardy–Weinberg equilibrium; 95% CI, 95% confidence interval; NA, not applicable. * Adjusted by age. *p*-values < 0.05 are bold.

**Table 3 ijms-26-06585-t003:** Allele combination analyses for *PCSK5* and *MUC2* polymorphisms in RPL patients and controls.

Allele Combinations	Controls	RPL	OR (95% CI)	*p*
	(2*n* = 472)	(2*n* = 676)		
*PCSK5* rs1110222 G > A/*PCSK5* rs2259969 A > G/*MUC2* rs7103978 A > G/*MUC2* rs10902088 C > T *				
G-A-A-C	224 (47.5)	308 (45.6)	1.000 (reference)	
G-A-A-T	120 (25.4)	208 (30.8)	1.261 (0.950–1.673)	0.106
G-A-G-C	8 (1.7)	0 (0.0)	**0.043 (0.003–0.746)**	**0.001**
G-A-G-T	12 (2.5)	26 (3.8)	1.576 (0.778–3.190)	0.179
G-G-A-C	19 (4.0)	23 (3.4)	0.880 (0.468–1.656)	0.694
G-G-A-T	13 (2.8)	19 (2.8)	1.063 (0.514–2.197)	0.869
G-G-G-C	2 (0.4)	4 (0.6)	1.455 (0.264–8.014)	1.000
G-G-G-T	0 (0.0)	0 (0.0)	-	-
A-A-A-C	10 (2.1)	13 (1.9)	0.945 (0.407–2.195)	0.897
A-A-A-T	23 (4.9)	9 (1.3)	**0.285 (0.129–0.627)**	**0.0003**
A-A-G-C	1 (0.2)	0 (0.0)	0.243 (0.010–5.987)	0.422
A-G-A-C	14 (3.0)	34 (5.0)	1.766 (0.926–3.369)	0.061
A-G-A-T	18 (3.8)	26 (3.8)	1.051 (0.562–1.963)	0.877
A-G-G-C	1 (0.2)	0 (0.0)	0.243 (0.010–5.987)	0.422
A-G-G-T	7 (1.5)	6 (0.9)	0.623 (0.207–1.880)	0.401
*PCSK5* rs1110222 G > A/*MUC2* rs7103978 A > G/*MUC2* rs10902088 C > T *				
G-A-C	243 (51.5)	332 (49.1)	1.000 (reference)	
G-A-T	133 (28.2)	227 (33.6)	1.249 (0.953–1.637)	0.104
G-G-C	10 (2.1)	2 (0.3)	**0.146 (0.032–0.674)**	**0.006**
G-G-T	12 (2.5)	27 (4.0)	1.647 (0.818–3.316)	0.134
A-A-C	24 (5.1)	46 (6.8)	1.403 (0.834–2.361)	0.186
A-A-T	41 (8.7)	35 (5.2)	0.625 (0.386–1.010)	0.054
A-G-C	3 (0.6)	2 (0.3)	0.488 (0.081–2.944)	0.655
A-G-T	6 (1.3)	5 (0.7)	0.610 (0.184–2.022)	0.541
*PCSK5* rs1110222 G > A/*MUC2* rs10902088 C > T *				
G-C	253 (53.6)	334 (49.4)	1.000 (reference)	
G-T	145 (30.7)	254 (37.6)	**1.327 (1.022–1.723)**	**0.032**
A-C	26 (5.5)	48 (7.1)	1.398 (0.844–2.316)	0.178
A-T	48 (10.2)	40 (5.9)	**0.631 (0.402–0.990)**	**0.044**
*PCSK5* rs1110222 G > A/*PCSK5* rs2259969 A > G				
G-A	364 (77.2)	541 (80.1)	1.000 (reference)	
G-G	34 (7.2)	47 (6.9)	0.930 (0.587–1.475)	0.758
A-A	34 (7.2)	23 (3.3)	**0.455 (0.264–0.786)**	**0.004**
A-G	40 (8.5)	66 (9.7)	1.110 (0.733–1.681)	0.621
*MUC2* rs7103978 A > G/*MUC2* rs10902088 C > T				
A-C	266 (56.5)	378 (55.9)	1.000 (reference)	
A-T	175 (37.0)	262 (38.8)	1.054 (0.823–1.349)	0.680
G-C	13 (2.7)	4 (0.6)	**0.217 (0.070–0.672)**	**0.005**
G-T	18 (3.9)	32 (4.7)	1.251 (0.688–2.276)	0.462

PCSK5, proprotein convertase subtilisin/kexin type 5; MUC2, mucin 2; OR, odds ratio; 95% CI, 95% confidence interval. *p*-values < 0.05 are bold. * Allele combination types were calculated by MDR method.

**Table 4 ijms-26-06585-t004:** Genotype combination analyses for *PCSK5* and *MUC2* polymorphisms in RPL patients and controls.

Genotype Combinations	Controls	RPL	AOR (95% CI)	*p **
	(*n* = 236)	(*n* = 338)		
*PCSK5* rs1110222 G > A/*PCSK5* rs2259969 A > G				
GG/AA	140 (59.3)	223 (66.0)	1.000 (reference)	
GG/AG	23 (9.7)	30 (8.9)	0.801 (0.444–1.447)	0.462
GG/GG	3 (1.3)	6 (1.8)	1.205 (0.296–4.915)	0.794
GA/AA	26 (11.0)	14 (4.1)	**0.335 (0.169–0.664)**	**0.002**
GA/AG	38 (16.1)	53 (15.7)	0.849 (0.531–1.359)	0.496
GA/GG	2 (0.8)	3 (0.9)	0.922 (0.152–5.597)	0.930
AA/AA	2 (0.8)	2 (0.6)	0.615 (0.085–4.430)	0.629
AA/AG	1 (0.4)	3 (0.9)	1.788 (0.183–17.464)	0.617
AA/GG	1 (0.4)	4 (1.2)	2.446 (0.270–22.137)	0.426
*PCSK5* rs1110222 G > A/*MUC2* rs7103978 A > G				
GG/AA	147 (62.3)	234 (69.2)	1.000 (reference)	
GG/AG	19 (8.1)	25 (7.4)	0.818 (0.434–1.539)	0.533
GA/AA	55 (23.3)	59 (17.5)	**0.653 (0.427–0.998)**	**0.049**
GA/AG	11 (4.7)	11 (3.3)	0.634 (0.268–1.500)	0.299
AA/AA	3 (1.3)	9 (2.7)	1.827 (0.486–6.873)	0.373
AA/AG	1 (0.4)	0 (0.0)	-	0.993
*PCSK5* rs1110222 G > A/*MUC2* rs10902088 C > T				
GG/CC	69 (29.2)	83 (24.6)	1.000 (reference)	
GG/CT	71 (30.1)	127 (37.6)	1.490 (0.967–2.295)	0.071
GG/TT	26 (11.0)	49 (14.5)	1.581 (0.890–2.807)	0.118
GA/CC	14 (5.9)	21 (6.2)	1.209 (0.569–2.569)	0.621
GA/CT	39 (16.5)	38 (11.2)	0.807 (0.463–1.407)	0.450
GA/TT	13 (5.5)	11 (3.3)	0.675 (0.283–1.610)	0.376
AA/CC	0 (0.0)	4 (1.2)	-	0.995
AA/CT	3 (1.3)	1 (0.3)	0.230 (0.023–2.324)	0.213
AA/TT	1 (0.4)	4 (1.2)	3.165 (0.343–29.214)	0.310
*PCSK5* rs2259969 A > G/*MUC2* rs7103978 A > G				
AA/AA	150 (63.6)	217 (64.2)	1.000 (reference)	
AA/AG	18 (7.6)	22 (6.5)	0.844 (0.437–1.627)	0.612
AG/AA	49 (20.8)	75 (22.2)	1.024 (0.673–1.558)	0.912
AG/AG	13 (5.5)	11 (3.3)	0.578 (0.252–1.329)	0.197
GG/AA	6 (2.5)	10 (3.0)	1.116 (0.396–3.148)	0.835
GG/AG	0 (0.0)	3 (0.9)	-	0.994
*PCSK5* rs2259969 A > G/*MUC2* rs10902088 C > T				
AA/CC	65 (27.5)	78 (23.1)	1.000 (reference)	
AA/CT	75 (31.8)	115 (34.0)	1.284 (0.827–1.993)	0.266
AA/TT	28 (11.9)	46 (13.6)	1.368 (0.770–2.430)	0.285
AG/CC	16 (6.8)	25 (7.4)	1.242 (0.605–2.549)	0.556
AG/CT	34 (14.4)	46 (13.6)	1.100 (0.632–1.914)	0.737
AG/TT	12 (5.1)	15 (4.4)	1.000 (0.434–2.303)	0.999
GG/CC	2 (0.8)	5 (1.5)	2.096 (0.393–11.174)	0.386
GG/CT	4 (1.7)	5 (1.5)	0.991 (0.254–3.865)	0.990
GG/TT	0 (0.0)	3 (0.9)	-	0.994
*MUC2* rs7103978 A > G/*MUC2* rs10902088 C > T				
AA/CC	76 (32.2)	106 (31.4)	1.000 (reference)	
AA/CT	95 (40.3)	141 (41.7)	1.069 (0.720–1.586)	0.742
AA/TT	34 (14.4)	55 (16.3)	1.162 (0.691–1.953)	0.572
AG/CC	7 (3.0)	2 (0.6)	**0.193 (0.039–0.960)**	**0.045**
AG/CT	18 (7.6)	25 (7.4)	0.998 (0.508–1.958)	0.995
AG/TT	6 (2.5)	9 (2.7)	1.076 (0.367–3.159)	0.893

PCSK5, proprotein convertase subtilisin/kexin type 5; MUC2, mucin 2; RPL, recurrent pregnancy loss; AOR, adjusted odds ratio; 95% CI, 95% confidence interval. * Adjusted by age. *p*-values < 0.05 are bold.

## Data Availability

The data presented in this study can be made available upon request by the corresponding author.

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
