# Peer review of "Genetic Association of PCSK5 and MUC2 Gene Polymorphisms with Recurrent Pregnancy Loss (RPL)"

_ijms, 2025, doi:10.3390/ijms26146585_

Round 1

Reviewer 1 Report

Comments and Suggestions for Authors

This study investigates recurrent pregnancy loss and aims to evaluate whether risk factors previously identified in the context of endometriosis could also be relevant in RPL. Furthermore, the authors aim to focus specifically on single nucleotide variants present in Korean women, a population that is currently understudied in this context.

The study identifies several variants that are associated with increased or decreased risk of RPL, which therefore have the potential to become significant biomarkers to diagnose early predisposition. They study is interesting and well-written. I only have a few points for the authors:

Major comments: 

1) The authors provide several baseline characteristics between recurrent pregnancy loss patients and controls. Is it possible to relate whether there is an association between the variants identified and alterations in these baseline characteristics?

2) Do the authors know whether age of the subjects is an additional variable influencing whether these single nucleotide variant pose a risk in the context of RPL?

Minor comments:

1) Table 2 should be fixed and formatted properly, at the moment it is difficult to read and therefore, appreciate.

2) There are references in the introduction that are not formatted correctly

3) The graphs should include the exact number of patients in the legend

Author Response

Response to the Reviewer’s comments

Reviewer 1

Major comments: 

1) The authors provide several baseline characteristics between recurrent pregnancy loss patients and controls. Is it possible to relate whether there is an association between the variants identified and alterations in these baseline characteristics?

-> We thank the reviewer for the valuable comments. Unfortunately, we are trying to find the association, but we have not identified it. However, we find an association between these genes and reproductive disorders, including pregnancy loss. The mentioned paragraph is as follows: “Recent bioinformatic analyses have identified PCSK5 as a potential biomarker in RPL [33]. Additionally, altered MUC2 gene expression has been observed in the endometrium of patients with recurrent implantation failure [34].”

2) Do the authors know whether age of the subjects is an additional variable influencing whether these single nucleotide variant pose a risk in the context of RPL?

-> We thank the reviewer for the critical comments. As you mentioned, we know the association between age and recurrent pregnancy loss. So, we compared the controls and RPL patients in Table 1. Since there was no significant difference between the controls and RPL patients, we proceeded with further study.

Minor comments:

1) Table 2 should be fixed and formatted properly, at the moment it is difficult to read and therefore, appreciate.

-> We thank the reviewer for the valuable comment. As you mentioned, we fixed Table 2.

2) There are references in the introduction that are not formatted correctly

-> We thank the reviewer for the valuable comment. As you mentioned, we revised the references in the introduction. The revised sentence is as follows: “Recent research has demonstrated the potential to detect pathogenic and exonic RPL-associated variants using NGS [8,9].”

3) The graphs should include the exact number of patients in the legend

-> We thank the reviewer for the valuable comment. As you mentioned, we have added the number of patients to the figure legend. The revised legends are as follows: “Figure 3. CD4 helper T cell proportion disparities by MUC2 rs7103978 A>G polymorphisms by analysis of variance (ANOVA) in RPL patients. The MUC2 rs7103978 A>G polymorphisms have confirmed the association with increased CD4 helper T cell percentage [(A) AA vs. AG: AA, 35.81±6.78; AG, 49.25±8.66; P < 0.001. (B) A allele vs. G allele: A allele, 36.15±7.10; G allele, 49.25±8.66; P < 0.001; the number of patients: 81].”, “Figure 4. CD56 natural killer cell proportion disparities by MUC2 rs7103978 A>G polymorphisms by analysis of variance (ANOVA) in RPL patients. The MUC2 rs7103978 A>G polymorphisms have confirmed the association with decreased CD56 natural killer cell proportion [(A) AA vs. AG: AA, 18.13±7.60; AG, 14.08±5.05; P = 0.043. (B) A allele vs. G allele: A allele, 17.90±7.52; G allele, 14.08±5.04; P = 0.031; the number of patients: 139].”

Reviewer 2 Report

Comments and Suggestions for Authors

The paper addresses a specific and underexplored gap in reproductive genetics. The study responds to the gap where many RPL cases remain unexplained, particularly from a genetic perspective in Korean women. Based on the content of the article the authors should consider the following methodological improvements and further controls to strengthen the study:

  1. Only 45 women were sequenced using WES out of 574 total participants. Increase WES coverage or apply targeted resequencing in a larger subset to improve statistical power and detect rarer variants.
  2. The study only includes Korean women, limiting generalizability. Include diverse ethnic cohorts or replicate findings in different populations to assess whether associations are ethnicity-specific.
  3. No mention of controlling for environmental exposures, lifestyle, or dietary factors that could influence RPL. Adjust for confounding variables such as smoking, alcohol use, BMI changes, and stress levels.

In summary, while the study is methodologically solid with its prospective, controlled, and randomized design, integrating these improvements would significantly enhance the internal and external validity, reduce bias, and provide stronger evidence for clinical recommendations.

Author Response

Response to Reviewer’s comments

Reviewer 2

The paper addresses a specific and underexplored gap in reproductive genetics. The study responds to the gap where many RPL cases remain unexplained, particularly from a genetic perspective in Korean women. Based on the content of the article the authors should consider the following methodological improvements and further controls to strengthen the study:

  1. Only 45 women were sequenced using WES out of 574 total participants. Increase WES coverage or apply targeted resequencing in a larger subset to improve statistical power and detect rarer variants.

-> We thank the reviewer for the valuable comment. As you mentioned, we will further plan for applying to a larger subset. Additionally, we included sentences on the limitations of small sample sizes in WES results. The added sentence is as follows: “Fourth, the WES participants were only 45 women, so we will further progress by applying to a larger subset to improve statistical power and detect rare variants.”

  1. The study only includes Korean women, limiting generalizability. Include diverse ethnic cohorts or replicate findings in different populations to assess whether associations are ethnicity-specific.

-> We thank the reviewer for the valuable comment. As you mentioned, we plan to replicate our results in the other cohorts as well. Then, we already mentioned the limitations of our results in the discussion section. The mentioned sentences are as follows: “Third, the study population was limited to Korean women, which may restrict the generalizability of the findings. Further epidemiological and functional studies are needed to confirm and expand upon these results.”

  1. No mention of controlling for environmental exposures, lifestyle, or dietary factors that could influence RPL. Adjust for confounding variables such as smoking, alcohol use, BMI changes, and stress levels.

-> We thank the reviewer for the valuable comment. As you mentioned, we plan to investigate more specific confounding variables and attempt to adjust our analysis of the results accordingly. Additionally, we included sentences on the limitations of confounding variables in the discussion section. The added sentence is as follows: “Fifth, we had no adjusting confounding variables, so we will plan to investigate more specific confounding variables and attempt to adapt our analysis of the results.”

In summary, while the study is methodologically solid with its prospective, controlled, and randomized design, integrating these improvements would significantly enhance the internal and external validity, reduce bias, and provide stronger evidence for clinical recommendations.